# Enhancing Diagnostic Images to Improve the Performance of the Segment Anything Model in Medical Image Segmentation

**DOI:** 10.3390/bioengineering11030270

**Published:** 2024-03-09

**Authors:** Luoyi Kong, Mohan Huang, Lingfeng Zhang, Lawrence Wing Chi Chan

**Affiliations:** Department of Health Technology and Informatics, The Hong Kong Polytechnic University, Hong Kong SAR, China; luoyi.kong@connect.polyu.hk (L.K.); mo-han.huang@connect.polyu.hk (M.H.); 22067977g@connect.polyu.hk (L.Z.)

**Keywords:** medical image, computer-aided diagnosis systems, image enhancement, artificial intelligence algorithm

## Abstract

Medical imaging serves as a crucial tool in current cancer diagnosis. However, the quality of medical images is often compromised to minimize the potential risks associated with patient image acquisition. Computer-aided diagnosis systems have made significant advancements in recent years. These systems utilize computer algorithms to identify abnormal features in medical images, assisting radiologists in improving diagnostic accuracy and achieving consistency in image and disease interpretation. Importantly, the quality of medical images, as the target data, determines the achievable level of performance by artificial intelligence algorithms. However, the pixel value range of medical images differs from that of the digital images typically processed via artificial intelligence algorithms, and blindly incorporating such data for training can result in suboptimal algorithm performance. In this study, we propose a medical image-enhancement scheme that integrates generic digital image processing and medical image processing modules. This scheme aims to enhance medical image data by endowing them with high-contrast and smooth characteristics. We conducted experimental testing to demonstrate the effectiveness of this scheme in improving the performance of a medical image segmentation algorithm.

## 1. Introduction

Medical imaging plays a vital role in contemporary healthcare systems as an essential tool for assisting doctors in diagnosis by providing detailed information about the internal structures of the human body. Semantic segmentation, on the other hand, is a crucial field in artificial intelligence and an integral part of computer vision technologies. It aims to classify each pixel in a digital image to identify sets of pixels that constitute distinct categories. With the rapid advancement of computer technology in recent years, computer-aided diagnosis (CAD) has introduced numerous computer algorithms into the medical field. Semantic segmentation techniques, as significant algorithms within CAD systems, are widely applied in the medical field.

Unlike segmenting objects (e.g., animals and houses) from natural images, medical image segmentation requires higher accuracy. However, in practical testing, the accuracy of medical image segmentation tasks is often lower than that of traditional semantic segmentation tasks. The main reasons for this discrepancy lie in the quantity and quality of the datasets. Regarding quantity, medical images are associated with strong privacy concerns, resulting in only relatively small datasets being available. The commonly used semantic segmentation models have complex network structures, which means a strong modeling capacity. However, medical image datasets often have a limited data volume and simple features, making it easy for models to learn the features present in the data, even amid noise. This makes the models more prone to overfitting. In terms of image quality, the existing medical imaging techniques primarily include Computed Tomography (CT) and Magnetic Resonance Imaging (MRI), both of which may cause discomfort or risk to patients to various extents. CT scans involve the use of X-rays, which can be harmful to patients due to the presence of ionizing radiation. For example, Brenner et al.’s study [1] found that before 2007, approximately 0.4% of cancer occurrences were due to using CT scans. However, there is a possibility that this proportion could increase to a range of 1.5% to 2%. MRI scans are generally considered to be a safer alternative to CT scanning. However, its application is often limited in pediatric and emergency cases due to the requirement of patients to lie still for a longer duration [2]. Therefore, hospitals often employ low-dose or shorter imaging protocols for capturing medical images, significantly reducing patient risk or discomfort. However, this approach comes at the cost of compromising image quality. For instance, reducing the ionizing radiation exposure during a CT scan increases noise levels. Typically, when the radiation dose is reduced by 50%, the noise increases by 40% [3]. Medical image segmentation models trained on small datasets are often highly susceptible to interference from noise in low-quality medical images, ultimately affecting the segmentation accuracy [4].

Currently, an increasing number of researchers are exploring the integration of medical imaging into treatment processes to monitor therapy progress, plan treatment strategies, and predict prognoses. Park et al. [5] investigated the impact of incorporating MRI ancillary features into the Liver Imaging Reporting and Data System Treatment Response (LR-TR) algorithm to predict the pathological tumor response in local–regional therapy (LRT) of hepatocellular carcinoma (HCC). Accurate prediction of pathological tumor response enables hospitals to allocate donor livers more effectively. Otazo et al. [6] provided a comprehensive review of the development of MRI-guided radiation therapy, highlighting the revolutionary changes medical imaging has brought to cancer treatment by enabling more precise radiation therapy. Liu et al. [7] employed a deep learning algorithm, label-GAN, to synthesize synthetic dual-energy CT (sDECT) from magnetic resonance images, facilitating MRI-based treatment planning for proton therapy.

In this study, we primarily introduce generic digital image enhancement techniques combined with medical image processing techniques to enhance medical image quality and improve the accuracy of the medical image segmentation models. Generic digital image enhancement techniques are mainly employed for denoising and enhancing global contrast in images. In contrast, medical image processing techniques focus on targeted processing of specific regions of interest (such as the liver). In this article, we provide an overview of the medical image enhancement strategy, including its architecture, datasets, data-processing methods, detailed experimental setup, and performance evaluation metrics. The test results are comprehensively measured using eight metrics, including the Intersection over Union (IoU) and Dice coefficient.

In summary, the main contributions of this study are as follows:Digital image enhancement strategies for medical image features: Medical images often contain a high level of noise and details hidden in low-intensity regions. To address this, anisotropic diffusion filters are used for denoising, and histogram equalization is employed to enhance the texture and details of medical images globally. Additionally, the fusion of window level and window width adjustment techniques specific to medical images is applied to target the regions of interest within the image. These strategies aim to achieve global enhancement of medical images while highlighting the desired targets.Exploring the root causes of low accuracy in medical image segmentation and addressing them: The main reasons for the low accuracy in medical image segmentation are the limited quantity and low quality of medical image datasets. The Data Engine in the Segment Anything Model provides a continuous stream of data, thereby addressing the issue of scarce medical image data. This article attempts to use image-enhancement techniques to provide high-quality data to the model and, thus, improve its accuracy.

Finally, the following sections will provide a detailed overview of existing works on image enhancement and image segmentation, describe the implementation process of the proposed image-enhancement method, analyze its performance in actual medical image segmentation tasks, and provide a summary and discussion of the analysis results.

## 2. Related Works

### 2.1. Image Enhancement

Image-enhancement techniques purposefully enhance or suppress the overall or local features of an image to improve its visual representation for specific tasks [8]. Traditional image enhancement methods can be categorized into spatial and frequency domain transformations for digital images [9]. Spatial domain transformations directly process the pixels (or image elements) of an image. For example, in the research by Chang and Wu [10], contrast enhancement of X-ray images was achieved by formulating contrast gain as a function of local standard deviation for histogram transformation. Srinivasan et al. [11] used the associated histogram equalization technique to improve the edges of cervical images and, thus, aid in accurately detecting cervical cancer using neural networks. Lin et al. [4] proposed a novel adaptive unsharp masking filter (UMF) architecture that adjusted the gain of sharpening enhancement by constructing a hyperbolic tangent function, ultimately enhancing natural images captured under poor illumination conditions. Rajan et al. [12], with UMF as the core, proposed a preprocessing method for retinal optical coherence tomography (OCT) images, paving the way for subsequent segmentation and classification of OCT images. Frequency domain transformations decompose the variations of pixel values in an image into a linear combination of simple sinusoidal functions in terms of amplitude, spatial frequency, and phase. Specific frequency domain filtering is applied, and the transformed image is then inversely transformed to the spatial domain. There are various frequency domain processing methods, such as those based on Fourier transform (FT) [13,14,15], discrete cosine transform (DCT) [16,17,18], and discrete wavelet transform (DWT) [19,20,21]. Additionally, many researchers have explored novel image-enhancement methods by combining spatial domain transformations with frequency domain transformations [22,23].

### 2.2. Image Segmentation

Semantic segmentation of images is an important branch of computer vision. It associates image pixels with labels or categories and classifies them based on their pixel-wise, edge-wise, or region-wise characteristics to obtain a collection of pixels representing distinct categories. Fully Convolutional Networks (FCNs) have been a pioneering work in deep learning for semantic segmentation [24]. FCNs replace the fully connected layers in traditional Convolutional Neural Networks (CNNs) with convolutional layers and use deconvolutional layers to restore the feature maps to the original image size. Due to the lower resolution of the masks obtained by FCN networks, Badrinarayanan et al. introduced SegNet by adding pooling layers to the FCN architecture [25]. SegNet follows a completely symmetric structure consisting of convolution–deconvolution and pooling–unpooling layers, forming an encoder–decoder structure. However, both FCN and SegNet classify pixels independently without considering the correlations between pixels, making them less suitable for medical image segmentation. U-Net, proposed specifically for medical image segmentation, addresses this issue [26].. It extracts shallow and deep features of medical images through a U-shaped network structure and merges these features using skip connections, enabling more precise segmentation of medical images. Numerous researchers have improved the U-Net network, and the U-Net family of networks is widely recognized as the most suitable algorithm for medical image segmentation tasks [27,28,29,30,31]. However, current improvements seem to have reached a bottleneck, as even incorporating the Transformer architecture into U-Net does not achieve impressive performance in medical image segmentation, especially in tumor-segmentation tasks. Recently, the Segment Anything Model (SAM) [32], developed by Meta, has achieved high-accuracy semantic segmentation across almost all domains, demonstrating its powerful segmentation capability. Many researchers have also explored the performance of the SAM model in the medical field [33,34,35,36], and experimental results indicate its great potential in medical applications. However, it should be noted that SAM is not a one-size-fits-all solution, as pointed out by Chen et al. [37]. Adjustments to the model are necessary to adapt to camouflaged images, images with shadows, medical images, and other images.

In summary, existing image-enhancement methods fail to cater to the specific characteristics of medical images. Incorporating prior knowledge of medical imaging into the image enhancement process while considering the task-specific features of medical images can significantly enhance their practical value. This study aims to validate this hypothesis by focusing on medical image segmentation.

## 3. Methodology

### 3.1. Medical Image Enhancement Scheme—Procedure and Structure

The pixel value range of CT images is typically [−1024, 3071] Hounsfield Units (HU), which is based on the absorption level of water and reflects the X-ray absorption characteristics of different human tissues, with bone cortical tissue and air having upper and lower limits of HU values, respectively. On the other hand, the commonly used pixel value range for digital images is typically 8-bit, with a range of [0, 255]. Since the SAM model only supports 8-bit digital images as input data, medical images must be converted to the 8-bit format when using SAM for processing. During this conversion process, there may be a loss of contrast between organs or between organs and tumors. This is because mapping the pixel values of CT images to an 8-bit digital image significantly reduces the pixel differences between different tissues, resulting in a decrease in the overall image contrast.

This study designed a preprocessing scheme (see in Figure 1) specifically tailored to medical image features. It consists of two processing paths: global image enhancement and local tissue image enhancement. By separately processing the medical image globally and locally and merging the results, the scheme achieves noise reduction, contrast enhancement, and emphasis on target tissues, which are beneficial for subsequent medical image processing.

Various methods are available for image denoising, such as Gaussian filtering, mean filtering, and median filtering. However, most smoothing filters blur the edges while reducing noise, as seen in Gaussian and mean filtering, and edge features are crucial in almost all image-related tasks. Although median filtering effectively removes salt-and-pepper noise, it is not particularly suitable for medical images, which typically do not exhibit prominent salt-and-pepper noise. Taking these factors into consideration, the proposed method employs anisotropic diffusion, a denoising method that preserves edges, as seen in Figure 2.

There are two main methods commonly used for contrast enhancement: gray-level transformation and histogram transformation. Gray-level transformation involves mapping the pixel values of the original image to achieve enhanced contrast. On the other hand, histogram transformation directly alters the pixel distribution of the image to enhance contrast. It is particularly useful for images where both the background and foreground are either too bright or too dark. This method can significantly improve the display of tissue and organ structures in X-ray images and enhance details in overexposed or underexposed photographs (see Figure 3). Therefore, this study employs histogram equalization to enhance the global contrast of CT images.

### 3.2. Anisotropic Diffusion

In the research of medical image segmentation, achieving high contrast in the target tissue region is often crucial. Firstly, anisotropic diffusion filtering is applied to the original CT image to preliminarily eliminate image noise. Anisotropic diffusion filtering is primarily used for image smoothing [38], overcoming the drawbacks of Gaussian blurring. It preserves image edges while smoothing the image, making it widely used in image processing and computer vision to maintain detailed image features while reducing noise. The main iterative equation of anisotropic diffusion is as follows:(1)$$I_{t+1}=I_{t}+\lambda (cN_{x,y}\nabla_{N}(I_{t})+cS_{x,y}\nabla_{S}(I_{t})+cE_{x,y}\nabla_{E}(I_{t})+cW_{x,y}\nabla_{W}(I_{t}))$$
where I represents the image; t represents the number of iterations; and N, S, E, and W represent the north, south, east, and west directions, respectively. cN, cS, cE, and cW represent the diffusion coefficients in the four directions. The formula for anisotropic diffusion is as follows:(2)$$cN_{x,y}=\exp(-{\left\|\nabla_{N}(I)\right\|}^{2}/k^{2})$$
(3)$$cS_{x,y}=\exp(-{\left\|\nabla_{S}(I)\right\|}^{2}/k^{2})$$
(4)$$cE_{x,y}=\exp(-{\left\|\nabla_{E}(I)\right\|}^{2}/k^{2})$$
(5)$$cW_{x,y}=\exp(-{\left\|\nabla_{W}(I)\right\|}^{2}/k^{2})$$

The two parameters, *k* and *λ*, in anisotropic diffusion control the level of smoothing, with larger values resulting in smoother images but with the edges less preserved.

### 3.3. Histogram Equalization

Histogram equalization is a classical algorithm in digital image processing. It enhances the contrast of an image by mapping the pixel value histogram distribution to an approximately uniform distribution. This technique allows for richer brightness and color details in the image, revealing hidden information. It has applications in various fields, including medical image processing. The histogram-equalization process is closely related to the probability distribution of image pixels. First, the grayscale distribution of an image can be represented as
(6)$$h(r_{k})=n_{k}$$
where $n_{k}$ represents the number of pixels in the image with a grayscale value of $r_{k}$, where $r_{k}$ is the kth gray level with k=0,…,255. Thus, the probability density of the image histogram can be represented as
(7)$$P(r_{k})=\frac{h(r_{k})}{N}$$
where $n_{k}$ represents the number of pixels at the current gray level, N represents the total number of pixels in the image, and $P\left(r_{k}\right)$ represents the proportion of the total number of pixels in the image that have the *k*-th grayscale level. Then, by computing the cumulative probability density of the original image grayscale levels, we can obtain the probability density of the new image grayscale levels, which can be expressed as
(8)$$s_{k}=\sum_{j=0}^{k}P(r_{j}),k=0,1,\ldots ,255$$

Finally, the new probability density obtained is multiplied by the maximum grayscale value L of the image to obtain the pixel values of the image after histogram equalization. Pixel values $p_{k}$ of the image are updated according to
(9)$$p_{k}\leftarrow s_{k}\times (L-1)$$

### 3.4. Window Adjustment

Medical images represent the internal structures and functions of anatomical regions, presented in the form of two-dimensional pixels or three-dimensional voxels. There are six main formats for radiological images, including DICOM (Digital Imaging and Communications in Medicine), NIFTI (Neuroimaging Informatics Technology Initiative), PAR/REC (Philips MRI scan format), and ANALYZE (Mayo Medical Imaging), as well as NRRD (Nearly Raw Raster Data) and MINC (Medical Imaging NetCDF) formats. Among them, DICOM and NIFTI are the most commonly used formats. Taking DICOM as an example, the HU value can be defined as follows:(10)HU=pixel×slope+intercept
where pixel represents the pixel value (gray level), and slope and intercept are defined by the window level and window width:(11)$$slope=\frac{g}{w}$$
(12)$$intercept=(\frac{w}{2}-c)\times \frac{g}{w}$$
where *w* represents the window width, *c* represents the window level, and *g* represents the width of the mapping range (with a width of 255 for the range of [0, 255]). Therefore, the adjustment of window level and window width can be defined as
(13)$$\mathrm{WindowLeveling}(x)=\left\{\begin{array}{l} 0,x<c-\frac{w}{2}\\ \frac{g}{w}\times x+(\frac{w}{2}-c)\times \frac{g}{w},c-\frac{w}{2}\leq x\leq c+\frac{w}{2}\\ 255,x>c+\frac{w}{2} \end{array}\right.$$

## 4. Experiments and Results

### 4.1. Server Information

This study conducted model training and validation on a server equipped with the devices and OS shown in Table 1.

### 4.2. Model Selection

This study used the SAM-based model as a testing model to validate the effectiveness of the proposed preprocessing architecture. However, as pointed out by Chen et al. [37], SAM is not universally applicable, and its performance on medical images is not as promising as on traditional images. To explore the preprocessing scheme’s assistance in improving accuracy, we desired a model that could adapt to medical images well. Therefore, in this research, we employed MedSAM [34], a SAM-based model specifically fine-tuned for medical images, as the testing model.

### 4.3. Dataset

This study utilized the publicly available LiTS17 dataset [39], licensed under a Creative Commons Attribution-NonCommercial-NoDerivatives 4.0 International License. It comprises 130 CT scan cases. The data and segmentation masks were provided by seven clinical centers worldwide. As the provided masks in this dataset are for liver and liver tumor regions, this experiment showcased the usage and specific workflow of preprocessing approaches for the liver as the target region.

### 4.4. Evaluation Metrics

To objectively and comprehensively compare the impact of different preprocessing approaches on image segmentation, this study selected eight commonly used metrics in the field of image segmentation to quantitatively evaluate the segmentation results. Table 2 provides a brief introduction to these eight evaluation metrics.

### 4.5. Time Complexity Analysis

Time complexity is used to measure the extent to which the execution time of an algorithm increases with the growth of input size. It is one of the most important metrics for evaluating algorithm efficiency. In the proposed image-enhancement method, three branches are computed and analyzed for their time complexity.

Anisotropic diffusion

The anisotropic diffusion algorithm used in this study is an iterative method, where each iteration involves computing over local regions of the image. Assuming the image size is N×N and the number of iterations is T, the computational cost per iteration is $O\left(1\right)$. In this case, the algorithm typically requires traversing each image pixel to calculate the image’s gradient and construct the guiding function, resulting in a time complexity of $O\left(N^{2}\right)$ for this step. With *T* iterations, the overall time complexity of the algorithm can be expressed as $O\left(T\times N^{2}\right)$.

2.Histogram Equalization

Histogram equalization typically involves four steps, assuming the image size is N×N. Firstly, computing the image’s histogram requires traversing the image, resulting in a time complexity of $O\left(N^{2}\right)$. Then, performing the cumulative operation on the histogram to calculate cumulative frequencies has a time complexity of $O\left(N\right)$. Next, computing the mapping function for pixel values based on the cumulative histogram has a time complexity of $O\left(N\right)$. Finally, adjusting the pixel values of the image according to the mapping function has a time complexity of $O\left(N^{2}\right)$. All in all, the time complexity of histogram equalization is $O\left(N^{2}\right)$.

3.Window Adjustment

The adjustment of the window level and window width in medical imaging involves only one step, which is traversing the image and adjusting the pixel values based on the window level and window width parameters. Assuming the image size is N×N, its time complexity is $O\left(N^{2}\right)$. Taking the maximum time complexity of the aforementioned branches, the time complexity of the proposed image-enhancement method is $O\left(N^{2}\right)$.

### 4.6. Experimental Results and Analysis

The step-by-step improvement in image quality is subjectively discernible to the naked eye. Figure 4 illustrates the intermediate result images of each key step in the proposed preprocessing workflow, clearly showcasing the variations in image quality at each stage.

Figure 5 displays the pixel value histogram distributions of each process image, with the vertical axis logarithmically scaled. The histograms provide a visual representation of the overall characteristics of each image. The pixel values of the standardized processed images are mainly concentrated between [100, 255], exhibiting an overall brighter image with a lack of dark details. Even with zoomed-in sections, it is challenging to discern the image’s details, as shown in Figure 6a. The pixel distribution of the histogram-equalized images is noticeably more balanced, effectively revealing the dark details of the image, as shown in Figure 5b. However, due to its global nature, it still exhibits unclear edges in the target region (liver), as seen in Figure 6b. The pixel values of the images adjusted through the window level and window width result in a significantly higher number of pixels, with a value of 0 compared to other gray levels greater than 0. This causes the overall image to appear darker, with an abundance of black noise and uneven edges, as depicted in Figure 5g and Figure 6c. However, satisfactory contrast is achieved for the liver region. The result presented is the blended image, which combines the advantages of histogram equalization and window adjustment. It maintains high contrast while enriching dark details, reducing noise, and providing smoother edges, as seen in Figure 6d.

Table 3 presents the performance of different liver segmentation image preprocessing methods using MedSAM. The proposed method achieves the highest scores in six out of eight evaluation metrics, only slightly lagging window adjustment in terms of recall and sensitivity.

The segmentation images provide a clearer representation of the differences in segmentation outcomes. Figure 7 and Figure 8 display the segmentation results of cases 9 and 10, respectively. It is visually apparent that histogram equalization produces the mask with the widest range of pixel values. This is because although histogram equalization enhances global contrast, it results in unclear boundaries for the liver region, posing difficulties for the algorithm in handling region edges. Window adjustment and the proposed method exhibit similar segmentation performance, with the main difference being how they handle region boundaries. However, the images processed through window adjustment are susceptible to noise interference, resulting in uneven region edges, which interfere with the algorithm and lead to subpar performance in the generated mask’s boundary. In contrast, the images processed by the proposed algorithm and used as input for MedSAM produce masks with smooth boundaries, as demonstrated in Figure 7.

## 5. Discussion

This study provides an effective preprocessing approach for medical imaging, offering clear and detail-rich input images for processing and analysis. Traditionally, two methods can be used to improve the capabilities of a model: model improvement and data processing. Most current research focuses on improving the model, such as by creating a series of enhancements for the U-Net model [26,27,28,29,30,31]. that modify the model architecture to achieve lower computational complexity, stronger robustness, and better alignment with the task objectives and datasets. However, regardless of how the model is improved to fit the data, the model’s computations are based on the input data. If the data quality is poor, the model is bound to produce unsatisfactory results. This means that the data determine the upper limit of the model’s performance, and model improvements can only approach this limit. In this study, we improve the model’s accuracy from the perspective of input data and achieve highly desirable results. Compared to histogram-equalization methods, the proposed approach demonstrates higher contrast in the target region (the liver), emphasizing the edge and texture features of the target area. Compared to window-adjustment techniques, the proposed method exhibits better control over noise levels and edge smoothness. The contrast between the target and background, as well as the clarity of target edges in medical image segmentation tasks, directly influence segmentation accuracy. The experimental results also demonstrate that histogram equalization with lower contrast performs worse in segmentation accuracy compared to higher-contrast window adjustment and the proposed method (IoU: −0.081; dice: −0.0502; accuracy: −0.0082; precision: 0.1033; recall: −0.0199; sensitivity: −0.0199; F1: 0.0502; specificity: 0.01). Furthermore, window adjustment, which is susceptible to edge-disturbing noise, also yields lower segmentation accuracy compared to the proposed method (IoU: −0.0345; dice: −0.024; accuracy: −0.0027; precision: −0.0355; recall: +0.0009; sensitivity: 0.0009; F1: −0.024; specificity: −0.0028).

## 6. Conclusions

In conclusion, our research provides a novel data-preprocessing method specifically designed for medical image preprocessing. By blending the results of global enhancement and local enhancement in a weighted manner, we achieved outstanding results with low noise and high contrast. The preprocessing effectiveness was evaluated through liver segmentation using MedSAM. The six evaluation metrics reached optimal values, proving the effectiveness of the proposed preprocessing approach. Among them, the advantages of IoU and dice coefficient scores were the most significant. This means that the mask obtained by preprocessing the data processed via the proposed method and then using MedSAM has the highest overlap with the ground truth regions and does not produce too many false-positive values. In medical image segmentation tasks, high segmentation accuracy is crucial, as blurry segmentation results are meaningless. The liver region mask obtained through the proposed preprocessing method and MedSAM achieved an accuracy of over 90%, making it highly suitable for practical application in hospital diagnosis processes. Overall, the results are promising, indicating that our research can assist doctors and benefit patients in the future.

## 7. Future Directions

Although the proposed method in this study combines the advantages of histogram-equalization and window-adjustment techniques, highlighting the target region while reducing image noise and allowing for weight adjustments to showcase background details according to the task requirements, it ultimately involves manually modeling and extracting image features. In future research, it would be worth exploring end-to-end networks as alternatives to these steps or incorporating artificial intelligence networks into the preprocessing stage. Scholars have already begun utilizing neural networks for image enhancement. In a study by Li et al. [40], the Retinex model was used to supplement lighting conditions in weakly illuminated images, enhancing their practical value. In research conducted by Qian et al. [41], weakly supervised learning was employed to eliminate artifacts, distortion, and overexposure in non-destructive testing (NDT) images. Similar to medical images, NDT images were acquired using X-ray scanning. This indicates the potential of employing neural network approaches in medical imaging as well.

In addition, the proposed preprocessing method is not only suitable for medical image segmentation scenarios but can also be applied to all medical image-related tasks, such as classification, recognition, localization, and detection, by adjusting the weight proportions. This means that it holds high potential for various applications. For example, it can be used in CAD systems to improve the quality of input images and enhance the accuracy of lesion localization, which is commonly used in CAD systems. Looking ahead, we hope this image-preprocessing technique can improve the accuracy of various AI tasks, further advance the development of CAD systems, and ultimately make a real impact on patients’ lives.

## Figures and Tables

**Figure 1 bioengineering-11-00270-f001:**
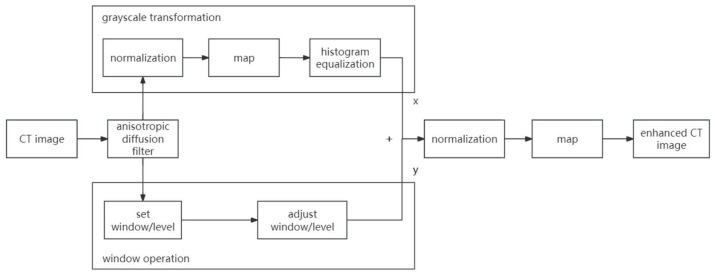
The proposed medical image preprocessing framework.

**Figure 2 bioengineering-11-00270-f002:**
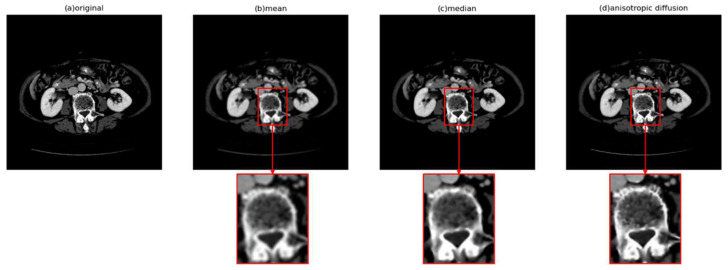
Visual effects of different denoising methods, where (**a**) is the original image.

**Figure 3 bioengineering-11-00270-f003:**
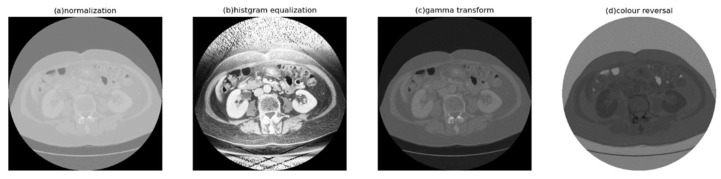
Visual effects of different contrast-enhancement methods, where (**a**) is the original image.

**Figure 4 bioengineering-11-00270-f004:**
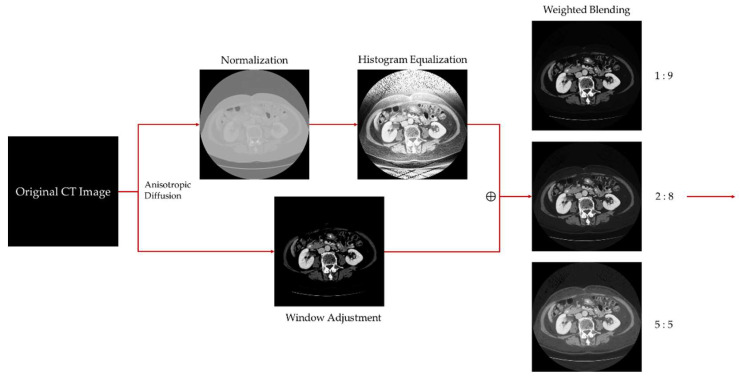
Intermediate results of each stage in image preprocessing.

**Figure 5 bioengineering-11-00270-f005:**
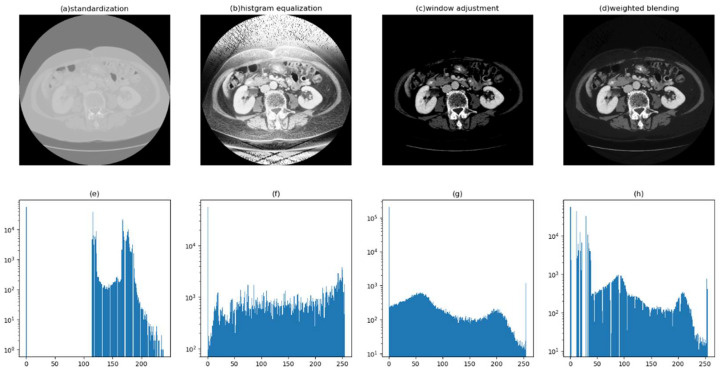
The relationship between the image’s visual effect and its corresponding histogram (shown below). (**a**–**d**) represent the process image of CT image preprocessing, while (**e**–**h**) correspond to the pixel histograms of the above images.

**Figure 6 bioengineering-11-00270-f006:**
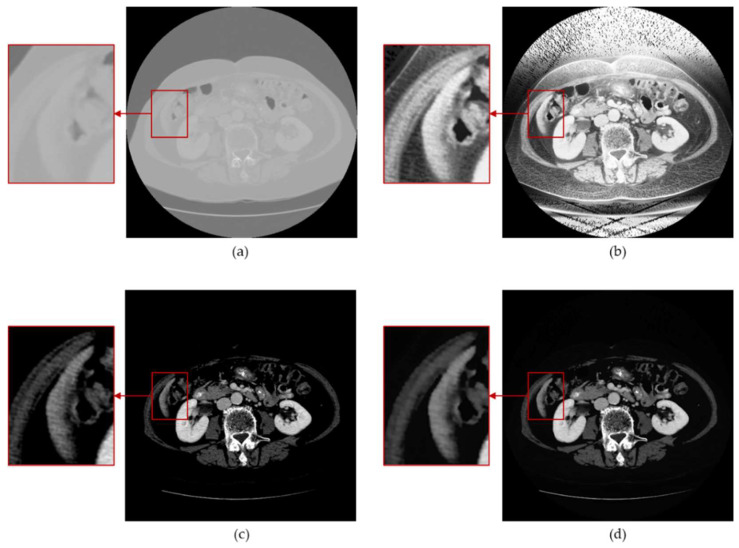
Zoomed-in detailed images of the preprocessing process. (**a**) normalization; (**b**) histogram equalization; (**c**) window adjustment; (**d**) proposed method.

**Figure 7 bioengineering-11-00270-f007:**
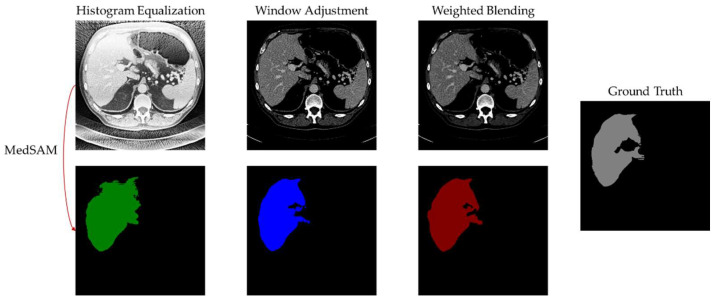
MedSAM segmentation result of case 9.

**Figure 8 bioengineering-11-00270-f008:**
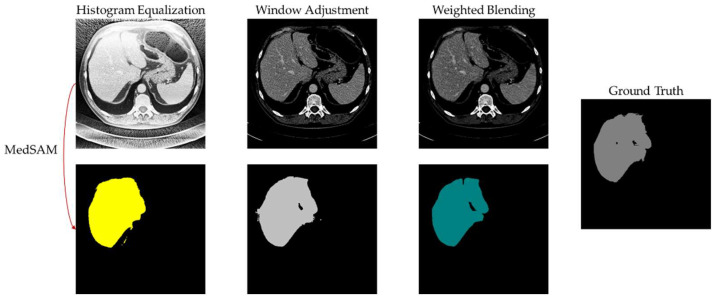
MedSAM segmentation result of case 10.

**Table 1 bioengineering-11-00270-t001:** Server configuration and environment.

Server configuration and Environment	
OS	Ubuntu 22.04.3 LTS
CPU	32 13th Gen Intel(R) Core(TM) i9-13900K
GPU	NVIDIA GeForce RTX 4090 × 2
RAM	32 GB DDR5 × 4

**Table 2 bioengineering-11-00270-t002:** The eight evaluation metrics used in the experiment.

Evaluation Metric	Description
IoU	Intersection over Union, which measures the overlap between the predicted and ground truth regions.
Dice Coefficient	Measures the similarity between the predicted and ground truth regions.
Accuracy	Measures the overall correctness of the segmentation.
Precision	Measures the proportion of correctly predicted positive pixels among all the predicted positive pixels.
Recall	Measures the proportion of correctly predicted positive pixels among all the ground truth positive pixels.
Sensitivity	Measures the ability to correctly identify positive pixels.
F1	Combines precision and recall to provide a balanced measure of segmentation accuracy.
Specificity	Measures the ability to correctly identify negative pixels.

**Table 3 bioengineering-11-00270-t003:** The scores of liver segmentation using MedSAM on medical images obtained using three different preprocessing methods.

	Proposed Method							
	IoU	dice	accuracy	precision	recall	sensitivity	f1	specificity
1	0.9635	0.9814	0.9970	0.9948	0.9683	0.9683	0.9814	0.9996
2	0.9164	0.9564	0.9902	0.9978	0.9182	0.9182	0.9564	0.9997
3	0.7682	0.8689	0.9988	0.8284	0.9135	0.9135	0.8689	0.9992
4	0.7752	0.8734	0.9923	0.8915	0.8559	0.8559	0.8734	0.9967
5	0.9009	0.9479	0.9854	0.9207	0.9766	0.9766	0.9479	0.9867
6	0.8816	0.9371	0.9846	0.9137	0.9616	0.9616	0.9371	0.9878
7	0.9367	0.9673	0.9920	0.9894	0.9462	0.9462	0.9673	0.9986
8	0.9428	0.9706	0.9908	0.9926	0.9495	0.9495	0.9706	0.9987
9	0.9577	0.9784	0.9976	0.9894	0.9676	0.9676	0.9784	0.9994
10	0.9594	0.9793	0.9933	0.9804	0.9782	0.9782	0.9793	0.9962
$\bar{x}$	**0.9002**	**0.9460**	**0.9922**	**0.9499**	0.9436	0.9436	**0.9460**	**0.9962**
	**Normalization**							
	IoU	dice	accuracy	precision	recall	sensitivity	f1	specificity
1	0.9576	0.9784	0.9965	0.986	0.9709	0.9709	0.9784	0.9988
2	0.8668	0.9286	0.9828	0.9053	0.9532	0.9532	0.9286	0.9867
3	0.6667	0.8000	0.9979	0.6732	0.9856	0.9856	0.8000	0.998
4	0.5641	0.7213	0.9767	0.5744	0.9693	0.9693	0.7213	0.9769
5	0.7928	0.8844	0.9661	0.8265	0.9511	0.9511	0.8844	0.9685
6	0.8652	0.9278	0.9816	0.8703	0.9933	0.9933	0.9278	0.9800
7	0.8661	0.9283	0.9821	0.9309	0.9256	0.9256	0.9283	0.9902
8	0.9315	0.9646	0.9885	0.9474	0.9824	0.9824	0.9646	0.9896
9	0.7801	0.8765	0.9854	0.8307	0.9276	0.9276	0.8765	0.9888
10	0.9010	0.9479	0.9826	0.9217	0.9758	0.9758	0.9479	0.9840
$\bar{x}$	0.8192	0.8958	0.984	0.8466	0.9635	0.9635	0.8958	0.9862
	**Window Adjustment**							
	IoU	dice	accuracy	precision	recall	sensitivity	f1	specificity
1	0.9605	0.9798	0.9968	0.9946	0.9655	0.9655	0.9798	0.9995
2	0.8886	0.9410	0.9869	0.9964	0.8914	0.8914	0.941	0.9996
3	0.7224	0.8389	0.9986	0.8454	0.8324	0.8324	0.8389	0.9994
4	0.5290	0.6919	0.9725	0.5314	0.9914	0.9914	0.6919	0.9719
5	0.8988	0.9467	0.9855	0.9506	0.9429	0.9429	0.9467	0.9923
6	0.8887	0.9411	0.9857	0.9246	0.9581	0.9581	0.9411	0.9895
7	0.9352	0.9665	0.9918	0.9902	0.9440	0.9440	0.9665	0.9987
8	0.9347	0.9662	0.9891	0.9515	0.9814	0.9814	0.9662	0.9905
9	0.9526	0.9757	0.9973	0.9939	0.9582	0.9582	0.9757	0.9997
10	0.9463	0.9724	0.9910	0.9657	0.9792	0.9792	0.9724	0.9933
$\bar{x}$	0.8657	0.922	0.9895	0.9144	**0.9445**	**0.9445**	0.9220	0.9934

## Data Availability

The data presented in this study are openly available in Codalab at https://competitions.codalab.org/competitions/17094 (accessed on 26 February 2024) or https://doi.org/10.1016/j.media.2022.102680 (accessed on 26 February 2024), reference number [39].

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
