# Peer review of "Enhancing Diagnostic Images to Improve the Performance of the Segment Anything Model in Medical Image Segmentation"

_bioengineering, 2024, doi:10.3390/bioengineering11030270_

Round 1

Reviewer 1 Report

Comments and Suggestions for Authors

The paper proposes a medical image enhancement scheme that integrates generic digital image processing and medical image processing modules. The paper is generally well-written and organized. However, some issues need to be solved:

1.       Abstract: the text in parentheses at the end of the last sentence is confusing in the context;

2.       Lines 37-38: the sentence “Unlike traditional semantic segmentation of digital images, medical image segmentation requires higher accuracy.” is inexact. There are many such “traditional” domains where image segmentation requires higher accuracy (e.g., remote sensing). Please reformulate;

3.       At the end of Section 1, please provide a brief description of the rest of the paper;

4.       Section 2 needs to be ended with a conclusion stating there is a need (a research gap) for this paper in the mentioned context of existing research works;

5.       The proposed method is general (e.g., both CT and MRI) or addressing only the CT case as in Figure 1?

6.       Some details regarding the selection of each algorithm/method to be included in the proposed overall methodology need to be added.

7.        Lines 255-256: the sequence “we desired a model that is well-adapted to medical image segmentation tasks” is confusing. Please explain;

8.       The title of Figure 4 needs to better specify its content.

9.       Required back matter sections (e.g. Author Contributions, Funding, etc.) need to be added at the end of the paper. Please see the journal’s Instructions for Authors.

10.   Reference list: Not all references are properly formatted. Please see the journal’s Instructions for Authors.

Reviewer 2 Report

Comments and Suggestions for Authors

I appreciate the opportunity to review the paper "" . This is an interesting study on medical image analysis in current cancer diagnosis. However, the quality of medical images is often compromised to minimise the potential risks associated with image acquisition by patients. Computer-aided diagnosis systems have advanced considerably in the last 11 years. These systems use computer algorithms to identify abnormal features in medical images, helping radiologists to improve diagnostic accuracy and achieve consistency in the interpretation of images and diseases. Importantly, the quality of medical images, as target data, determines the level of performance achievable by artificial intelligence algorithms. However, the range of pixel values of medical images differs from that of digital images typically processed by artificial intelligence algorithms, and blind incorporation of this data for training may result in suboptimal algorithm performance. In this study, a medical image enhancement scheme is proposed that integrates generic digital image processing and medical image processing modules. This scheme aims to enhance medical image data with high contrast and smoothness features. The results indicate that this procedure is effective in improving the performance of a medical image segmentation algorithm (Segment Anything Model).

In the following, I will make a number of suggestions for the improvement of the work:

1. The introduction should include more citations updated in the last five years, the authors present 38 citations, it is advisable to reach at least 40, of which 50% are updated in the last five years, it is advisable that it is at least 60% given the relevance and innovation of the subject matter addressed in the work. Long paragraphs without citation are also found.

2. At the end of the introduction section, the objectives of the work and the research questions should be included.

3. The methodology section should include the following sub-sections: participants, instruments, procedure and data analysis (the type of analysis to be applied to test the research questions should be included). It is recommended that this section be reworded.

4. The section entitled "Experiments" should be renamed "Results" and the results should be presented here according to the research questions formulated.

5. Also, in the results section, when testing the effectiveness of different data processing methods, fit indicators should be found, for example Good Fit algorithms through structural equations or Rand Index Fit.

6. Specifically, table 3 contains a lot of data that is not well visualised, I suggest splitting the information into three tables.

7. In the conclusions section, a sub-section should be included indicating the limitations of the study.

8. Even if the authors have used image banks, they should refer to the protection of the images as well as the prior consent of the participants.

9. Review the way of citing table and figure legends, in accordance with the rules of the journal.

10. Review the citations in the references section and adjust them to the journal's standards.

Reviewer 3 Report

Comments and Suggestions for Authors

- The paper proposes a medical image pre-processing pipeline that aims at enhancing the segmentation performance of  "Segment Anything Model" (SAM);

- The benchmarks presented indicate that the proposed strategy is effective; nonetheless, my recommendation goes along the clarification of a few points in the manuscript (details below);  

- The first paragraph of Sec. 1 ("Introduction") should be better supported in term of previous publish works; also, besides diagnosis, it should emphasized  the role of medical imaging in treatment procedures;  

- The argument concerning the lack of medical images datasets (lines 40-42) should somehow be rephrased, taking into account, for example, the amount of data already available in the public domain (as reported for example in https://www.nature.com/articles/s41467-024-44824-z);  

- SAM is reported to be developed by OpenAI (lines 137-138) but published material seems to point out to Meta; 

- The term "thermal conductivity" when referring  to "c" (line 178) should be reconsidered, to what seems to be a well acknowledged "diffusion coefficient";  

- Equations (2)-(5) are one (of the two) of the proposals of Peron and Malik... why was this one selected?   - In line 197, it is quoted that "k= 1, 255", and that should be re-analysed ("k=0, 255"?);  

- In lines 206-207, it should be clarified that L is the maximum grayscale level  of the bit depth in use (not necessarily of the image itself);

- Eq 9 should be re-written, as "new pixel value" sounds a rather vague concept (eg:  pixels' gray scale values r_k will be updated according to: r_k <-- s_k*L );

- The source of the image in Fig 2 should be quoted;

- The definition of g and w should appear right after Eq. 12;

- In Eq. 13: 

. the rationale adopted should be mentioned (linear  behavior from 0 to 255, in the range [c-w/2; c+w/2] );

.  the middle eq. holds: 0 for x=c-w/2 and g for x=c+w/2, what seems right; however in line 236 it is stated that g= 256, what should be revised (probably g= n_max-n_im)=255-0=255);  

- Concerning the metrics, are they all calculated pixel-wise? Or IoU based on areas / "bounding boxes"?

- About  Figure 3:

. it does not seem to be in full agreement with Fig 1, so clarification is needed; . details should be provided about the "Standardization" step; 

. how are the values of c and w selected on the "Window Leveling step"?

. can the blending step increase the image's bit-depth (eg 9 bits)? Case not, how is that prevented? 

. the ratios presented in the blending step (eg 1:9) refer to the weights of top and bottom images respectively?  

- Concerning  Table 3:

. numbers are presented with rather small font size, so reformatting is needed;

. more information is needed about "case"... which configurations / settings were considered?

. what were the values of  k(eqs 2-5), w and c (eq 13), and the blending proportions used?      

- Benchmarking: it is strongly suggested that, for the sake of generalization, the proposed method is assessed in a different dataset.

Comments on the Quality of English Language

- The document should be fully revised in order to avoid the repetition of the same / similar words in the same sentence (eg: line 37: "segmentation"; lines 62-64: "image" / "images" ).

Reviewer 4 Report

Comments and Suggestions for Authors

A very interesting manuscript proposing a medical image enhancement scheme that integrates generic digital image processing and medical image processing modules. The proposed scheme targets to enhance medical image data and therefore to increase segmentation performance. The authors provide experimental testing that demonstrates the effectiveness of the proposed model “Segment Anything Model”.

There are a few comments:

1.       Line: 42 ‘This makes the models more prone to overfitting”, please explain or add a reference here.

2.       Figure 1 is not referenced in the manuscript body

3.       Figure 2 for histogram equalization is not required, the capabilities of this technique are rather well known, if the authors wish to keep this image (for training purposes) then a similar figure should be added or the previous section for anisotropic diffusion

4.       Line 225, please spell the MNIC acronym as already have done with all the others.

5.       Nice to present the server characteristics, however there is not presented information for the performance of the method in terms of time or computational complexity requirements. The authors may provide such data especially in comparison with the other techniques.

Round 2

Reviewer 1 Report

Comments and Suggestions for Authors

The paper has been significantly improved. The authors solved all the issues I raised.

Comments on the Quality of English Language

English needs minor polishing.

Reviewer 3 Report

Comments and Suggestions for Authors

- I acknowledge that the issues previously raised   were properly addressed.

- At the editor discretion, I leave teh following suggestion for table 3:

. Head section centred ("Proposed Method"; "Standardization"; "Window Adjustment");

. Results are presentes with 4 digits precision.. are the statistical fluctuation compatible with those figures

. Is the table an embedded image? If so, care should be taken in terms of the text quality of the final version
